# Radiogenomics, Breast Cancer Diagnosis and Characterization: Current Status and Future Directions

**DOI:** 10.3390/mps5050078

**Published:** 2022-10-03

**Authors:** Francesca Gallivanone, Gloria Bertoli, Danilo Porro

**Affiliations:** Institute of Molecular Bioimaging and Physiology (IBFM-CNR), Via F.lli Cervi 93, 20054 Milan, Italy

**Keywords:** breast cancer, diagnosis, imaging, radiogenomic, PET/CT, MRI

## Abstract

Breast cancer (BC) is a heterogeneous disease, affecting millions of women every year. Early diagnosis is crucial to increasing survival. The clinical workup of BC diagnosis involves diagnostic imaging and bioptic characterization. In recent years, technical advances in image processing allowed for the application of advanced image analysis (radiomics) to clinical data. Furthermore, -omics technologies showed their potential in the characterization of BC. Combining information provided by radiomics with –omics data can be important to personalize diagnostic and therapeutic work up in a clinical context for the benefit of the patient. In this review, we analyzed the recent literature, highlighting innovative approaches to combine imaging and biochemical/biological data, with the aim of identifying recent advances in radiogenomics applied to BC. The results of radiogenomic studies are encouraging approaches in a clinical setting. Despite this, as radiogenomics is an emerging area, the optimal approach has to face technical limitations and needs to be applied to large cohorts including all the expression profiles currently available for BC subtypes (e.g., besides markers from transcriptomics, proteomics and miRNomics, also other non-coding RNA profiles).

## 1. Introduction 

Breast cancer (BC) is a complex disease and the second leading cause of cancer-associated death in women [1]. In 2020, 2.3 million women were diagnosed with BC and 685, 000 women died worldwide from BC [2]. Notably, the number of patients affected by BC is rapidly increasing, especially in developed countries, with an estimate of a further increase for delays in diagnosis and treatment due to the COVID-19 pandemic. Early detection is a key step for BC diagnosis in order to improve survival. In the last decades, imaging emerged as a powerful tool both for early detection and the characterization of BC as well as for the subsequent monitoring of therapy response (i.e., [3,4]). Technological advances in image technologies as well as in imaging processing approaches have also contributed to give a central role to different diagnostic imaging tools such as X-ray mammography, ultrasound, magnetic resonance imaging (MRI) and positron emission tomography (PET) for BC diagnosis (e.g., [5,6,7,8,9]). One of the major advantages of imaging, over the bioptical evaluation of a finite set of tumor tissue portions, is the possibility to spatially inspect the entire tumor over time, both in vivo and non-invasively. Even today, in a clinical context, reporting of medical images is mainly performed by qualitative visual assessment. In recent years, the advances in image processing methodologies, derived by increasingly powerful informatics resources, allowed for the exploration of the advantages of the quantitative analysis of medical images, with the aim of supporting and enhancing the diagnostic confidence in oncological diseases including BC [10]. New computer vision tools, developed in non-medical contexts, were adapted and explored for their application to medical images, in order to extract quantitative features from the images of an entire tumor capturing the overall image content hidden to the naked eye [11]. The hypothesis is that information hidden to the naked eye can reflect and reveal biologically relevant information, such as tumor heterogeneity, that can be responsible for different clinical outcomes or different responses to therapy. The new paradigm of radiomics is thus emerging as the high-throughput extraction of quantitative features from medical images (imaging biomarkers) to characterize the imaging phenotype as the in vivo expression of genotype of oncological diseases including BC (e.g., [12]). The collection of imaging biomarkers and their correlation with clinical information, molecular data and genomic or proteomic assays is supposed to assess the prognosis, to support clinical decisions, and to predict the response to therapy [13,14]. In particular, in an approach of personalized medicine, a great interest in the research community arose from the potential in mining and combining radiomic imaging features with genomic data for improving BC patient’s diagnosis and prognosis (e.g., [15]). 

In this brief review, we highlight the main advantages of using imaging diagnostic power and genetic/epigenetic profiles, alone or in combination, underlying the limits of each technique and their potential future applications. The review was focused on the relationship between imaging and biological markers that are already usable in a clinical context such as circulating miRNAs. Our intent was to provide an overview of the works that can potentially be more easily reproduced in a clinical context; this would transfer radiogenomics effectively in the clinical setting, using molecular BC data that can be obtained with minimally invasive methods, and using samples that can be easily collected and analyzed without significant costs. This would allow for the generalization of the results using a large cohort and ensuring the reliability of the results.

## 2. Biomarkers for BC

### 2.1. Biochemical Markers: Advantages and Limitations

The purpose of the biochemical and molecular analysis of a biopsy is to analyse the expression of specific molecules (DNA, RNA, small non-coding RNA, proteins) or cell types selected a priori [16]. The molecular characterization of a tumor could integrate all the information coming from the -omics profiles; indeed, it includes all the data on the changes of genes (genomic profile), mRNAs (transcriptomic profile), non-coding RNAs, and DNA modification (epigenomic profile), metabolism (metabolomic profile) and proteins (proteomic profile) between healthy tissues and tumor samples. The integration of the information coming from all these profiles, i.e., by machine learning approaches, allows for the selection of diagnostic biomarkers. If this method is applied to different tissues of the same tumor, selected on their grade of prognosis, a prognostic biomarker could be proposed. This approach towards the precision medicine takes into account the individual differences within the tumoral tissue, improving the application of personalized medicine tools [17]. The molecular characterization of the tumor provides additional information for tumor subtyping and could identify genetic aberrations that allow the clinicians to provide the patient with the best therapeutic option. In BC, the current molecular classification divides the tumor into five groups, namely luminal A, luminal B, ErbB2/Her2+, basal and normal-like [18]. Each of them is characterized by specific marker expression, and it is associated with a different prognosis [19]. Luminal A BC is an estrogen receptor (ER)-positive, progesterone receptor (PR)-positive, HER2-negative tumor, having a low level of proliferation marker Ki67. These make the tumor a slow growing cancer with a good prognosis. Luminal B BC is an ER positive, PR and HER2-negative or positive tumor, with high levels of Ki67, which makes it grow faster. This implies a worse prognosis compared to Luminal A. HER2-enriched or positive BC is ER-negative, PR-negative and HER2-positive cancer, growing faster than luminal BC and having a poor prognosis. It could be successfully treated with drugs targeting the HER2 protein. Triple negative BC or basal BC lacks ER, PR, and HER2 proteins’ expression, and this makes it one of the more aggressive subtypes [20]. All of the molecular characterizations need to be performed on bioptic samples of the primary BC. The molecular profiling of BC bioptic tissues, although easier to be performed at lower costs, cannot replace the imaging analysis, but could be an auxiliary method to be flanked to clinical classical diagnosis. Liquid biopsy is emerging as a promising and minimally invasive tool in precision medicine [21]. Indeed, it is able to provide an overview of primary and metastatic tumors at different times, also considering spatial and temporal tumor heterogeneity, requiring only by a blood sample. This kind of biopsy allows the identification of different types of molecules, such as circulating tumor DNA (ctDNA) (also called circulating free DNA, cfDNA), circulating tumor cells, and different circulating small RNA, such as microRNAs and long non-coding RNAs.
*Circulating tumor DNA (ctDNA) or circulating free DNA (cfDNA)* are small fragments released in the blood system from the primary tumor or metastatic cells. They are DNA fragments less than 500 bp in length and exhibit the same somatic alteration present in the tumor from which they originate, including point mutations, chromosomal rearrangements, copy number variations, and DNA methylations. As the amount of cfDNA is very small, the detection method used for their quantitation is mainly based on polymerase chain reaction (PCR) amplification and next-generation sequencing (NGS). These DNA fragments could be actively released via microvesicle release or the degradation of apoptotic and necrotic cancer cells [22]. From the original discovery of cfDNA in 1948, several papers have demonstrated that these DNA fragments originated from tumor cells undergoing genomic instability. In 1994, it was demonstrated that these small DNA showed the same specific genomic mutation of the primary tumor [23,24]. cfDNA obtained by plasma isolation at different time points could be helpful in the description of the natural course of cancer development before and after therapeutic treatment.*Circulating tumor cells (CTCs).* The presence of disseminating tumor cells is a common feature of solid cancer, such as BC. The detection of these cells is associated with poor outcomes at the level of both overall survival and disease-free survival in BC [25]. Disseminated tumor cells are usually isolated from a patient’s bone marrow, with an invasive technique that is not always accepted by the patients. CTCs are epithelial cells released by the primary tumor in the number of less than 100 cells per ml of peripheral blood. They are able to differentiate cancer patients from healthy subjects [26].*Non-coding RNAs (ncRNA).* ncRNAs are crucial regulators of gene expression and are strongly associated with BC. The large family of ncRNAs includes several regulatory RNAs, such as microRNA (miRNAs), long non-coding RNAs (lncRNA), and circular RNAs (circRNAs). miRNAs are small RNAs of 19–25 nucleotides able to regulates the mRNA profiles inside each cells; they could also be secreted in the microenvironment of the tumor as well as in body biofluids (blood, lacrimae, urine, etc.). The list of miRNAs have been deposited into miRBse database (https://www.mirbase.org, accessed on 1 July 2022) [27], which annotated more than 38,500 predicted miRNA sequences (release V22.1).

LncRNAs are 200 nucleotides base long RNAs able to regulate the expression of miRNAs, thus affecting the cellular mRNAs’ profile. They are annotated in LNCipedia containing more than 127,000 releases (July 2022) [28]. 

circRNAs are single-stranded RNAs that are more stable than other non-coding RNAs due to their form, and are formed by back splicing of coding RNAs. Many circRNAs have been found in different types of cancer, where they also regulate the expression of miRNAs, acting as miRNA sponges, or regulating the expression of parental genes. circRNA are annotated in an updated database called CircNet 2.0 [29]. Among the main circRNAs involved in cancer, hsa-circ-005505, hsa-circ-0007289 and hsa-circ-0058514 are involved in the regulation of metastasis of BC cells, while hsa-circ-0000479 or hsa-circ-0001783 have a role in carcinogenesis (for a review see [30]). Over 58,300 circRNAs, 15,500 lncRNAs, and more than 18,000 mRNAs have been found in human serum, mainly associated with exosomes [31], suggesting the existence of a complex regulatory network for non-coding RNA release, and intercellular and intra-tissue communication. 

### 2.2. BC imaging Biomarkers: From Standard Quantification to Radiomics

The term “biomarker” was conceived to identify biological molecules that can be used as an indicator of normal or pathological processes [32]. With the massive use of imaging and the widespread use of quantitative image processing methodologies, this term was extended to imaging: imaging biomarkers were defined as information that can be extracted from images to characterize normal or pathological processes [33]. 

Imaging biomarkers may be classified as qualitative or quantitative [34,35]. 

Diseases can be described and reported by nuclear medicine physicians and radiologists using semantic lexicon features. Semantic features are those that are commonly used in clinical practice as radiology lexicon to describe regions of interest, highlighting the presence of disease aggressive behavior, morphology, infiltration and metastatic capacity [12,14]. These features, even if used in clinical practice and of prognostic value, have several limitations, including the subjective expertise of the referring physicians in image interpretation and their limited human ability to capture subtle disease features and their temporal changes using the naked eye. Examples of qualitative semantic breast imaging descriptors which are typically used in BC diagnosis are the shape of the tumor and the appearance of the margin [36]. 

On the contrary, quantitative imaging biomarkers are numerical data that are extracted on images using standard or advanced computational approaches. Quantitative approaches to medical image analysis have been developed and implemented thanks to technical advances in imaging systems. The underlying hypothesis for image quantification is that image contrast, quantified by imaging metrics, reflect the pathophysiology of tissues. In order to obtain mineable images, image quantification requires a massive standardization of all processes to extract image metrics, from acquisition to image analysis and mining quantitative data. Proper international initiatives stimulated this important task (e.g., [37,38]). Image quantification workflow in oncology (Figure 1) requires the definition of Volume Of Interest (VOI), using proper segmentation strategies, so that quantitative information is related to the considered oncological lesion. The quantitative information extracted from VOI are known as imaging features [38]. At the beginning, VOI standard quantification methodologies were developed to overcome the limitation of the semantic description of pathologies. Standard imaging biomarkers include mean glucose consumption reflected by Standardized Uptake Value (SUV) in PET [39] or cellular arrangement within VOI evaluated by Apparent Diffusion Coefficient (ADC) in Diffusion Weighted MR Imaging (DWI-MR) [40]. 

In recent years, radiomics has emerged to overcome standard quantification approaches considering BC images as a source of a large amount of quantitative data, not perceivable with human eyes and extracted from images by applying complex mathematical methods, such as texture analysis. Regardless of the image modality, an optimal and standardized radiomics clinical workflow [41] starts with the definition of the study design, including the proper choice of protocol and the standardization of data collection from the acquisition of images and clinical data to the setting of data analysis. Even if great efforts are made in multicentric research studies for standardization of image acquisitions across different scanners and manufacturers, published radiomics studies still present a large variability in terms of image acquisition protocols, thus affecting the generalizability of the results. The basic workflow of a radiomic study is presented in Figure 1.

Concerning the definition of the clinical protocol, one of the major issues is related to sample size. Since advanced image analysis produces big data to be mined, the sample size of the cohort and feature selection methodologies have to be properly considered in order to avoid limitation of the statistical power of the radiomic analysis, potentially generating data over-fitting in the decision models [14,42]. The standardization of radiomic analysis involves a large number of radiomic workflows, starting from image acquisition to the setting of decision models. Once acquired, images have to be segmented in order to identify a VOI or sub-volumes of interest within the lesions to be analysed [43]. VOI segmentation is a complex task in medical imaging. A variety of segmentation approaches have been proposed, even for a single image modality (e.g., [44,45,46,47]), however, at present, no segmentation strategy can be considered as a standard method for tumor segmentation. Semi-automatic or automatic segmentation strategies are suggested in order to avoid inter- and intraobserver variability of manual segmentation [48,49].

On segmented VOIs, several Imaging Features (IFs) can be extracted by several different methods. Besides standard IFs extracted from a lesion as a whole and referred as “local intensity features”, advanced methods allow for the definition of features classified into a number of families with clear computational methods that include morphological and statistical features and filtering techniques [50]. A great effort by the IBSI initiative [50] allowed for the partial compensation of the initial lack of standardization in terms of nomenclature, definitions of features, and methods and software for radiomic feature calculation. Each of the methods used for feature extraction allows for the extraction of a huge amount of data for a single patient so, as to generate hypotheses from data, data reduction (also known as feature selection) is a crucial step. Feature selection can be performed using different approaches, and it is mandatory to avoid redundant information in data before modelling the endpoint of interest. Accurate, reproducible and not-redundant imaging features can be used for modelling the endpoint of interest, exploring the relationship of IFs with clinical outcome, clinical parameters or immunohistological data (e.g., [13,14]). A model is intended to select relevant information for the outcome of interest in order to be able to make predictions for new input data. In addition to training and testing a model, a validation on an independent dataset is currently required to clearly define the validity of the developed approach.

Different imaging methods are included in diagnostic workups of BC. In the last years, an increasing number of scientific works were dedicated to exploring the impact of radiomic biomarkers by using different image modalities. 

X-ray mammography is widely diffused as primary imaging methods for screening and early detection of BC [51]. From a clinical point of view, despite its role as an effective screening tool, X-ray mammography presents disadvantages for women with dense breasts or those that have had surgical interventions [52,53]. The use of ultrasound imaging (US) in conjunction with X-ray mammography was proposed to enhance screening performance; however, the US is characterized by a high rate of false positives. 

The possibility to extract standard metrics from X-ray mammography, such as mammographic breast density, was described in different works (e.g., [54]). In recent years, radiomics works were published dealing with the analysis of X-ray mammography by using advanced image acquisition techniques such as Contrast-Enhanced Mammography (CEM) [55] or contrast-enhanced spectral mammography (CESM) [56]. Son et al. [57] worked on synthetic mammography reconstructed from digital breast tomosynthesis (DBT) in order to predict molecular subtypes of BC using radiomics signatures. Interestingly, the patient cohort was consistent (higher than 300 pz), and a temporally independent validation cohort was provided. Despite this, it was demonstrated that the radiomic signature is only able to distinguish triple negative tumors with respect to other molecular subtypes.

Breast ultrasound (US) showed potential as an imaging tool for BC in conjunction to X-ray mammography and the use of advanced acquisition techniques, such as elastography, Doppler, or contrast-enhanced US (CEUS), which can provide information about tissue properties. US is routinely implemented in BC, but few radiomic studies have focused on this image modality mainly because of technical issues affecting US images (such as the large variability of image quality from both acquisition processes and operator experience as well as from the presence of noise and artifacts). 

In a pilot study, Qui X et al. [58] proposed a US-based radiomic nomogram for predicting presurgical axillary lymph node metastasis status in BC. Even if used on a limited cohort (less than 200 patients) and without an external validation, the results suggest that radiomic signatures and nomograms are promising in a clinical setting.

Guo X et al. [59] applied an innovative approach using deep learning (DL) radiomics in a multicentre study by building a DL model based on US features to predict the risk of axillary non-sentinel lymph node involvement. The model was built on a large cohort of patients (more than 900 patients) and underwent both an internal and an external validation, showing a good performance to be proposed as a simple preoperative tool to promote personalized axillary management of BC.

The breast MRI is currently a well-established method to be used in clinical practice for the diagnosis and preoperative staging of BC with the advantage of preventing patient exposure to radiation. The advantages of MRI are also related to the possibility of obtaining different information in a single examination by performing a multiparametric acquisition, combining several parameters from morphological information of T1 and T2-weighted images, and to functional information derived from dynamic contrast-enhanced MRI (DCE-MRI) or DWI-MR.

Different works were devoted in the last decades to exploring the use of radiomic IFs for different clinical issues. Recently, several studies have explored the application of advanced machine learning methods and deep learning to breast MRI (e.g., [60,61]) publishing a clinical–radiomics model combining a DCE-based radiomics signature and clinical data to predict complete response after neoadjuvant chemotherapy in patients with axillary lymph node metastasis. 

Considering the need to improve the performance of clinical imaging for BC diagnosis, whole body techniques such as PET/CT and PET/MRI are currently explored and widely used in clinical contexts to detect and characterize BC. These techniques, besides allowing a characterization of primary BC from both a morphological and a functional point of view, provide a full BC staging, with the evaluation of the axilla [61] and of distant metastasis [62], thus allowing for the better tailoring of the therapy procedure. From a quantitative point-of-view, it is expected that, with the use of hybrid PET/CT and PET/MRI, the combined modalities should be able to highlight the complementary tumor characteristics in order to maximize the information that could be extracted with radiomics. 

At present, an increasing number of research studies was dedicated to exploring the advantages of information provided by hybrid imaging, even if the majority of these works were dedicated to evaluating the relationship between radiomic features and immunohistochemical BC data (e.g., [41,63,64,65,66]).

## 3. Radiogenomics: Combining Molecular and Imaging Biomarkers for BC Characterization

The concept of radiogenomics broadens the purpose of radiomics: in radiogenomics, IFs are supposed to reflect the underlying molecular and genotypic basis of BC, thus characterizing the disease at a finer level, interfacing system biology and imaging. The correlation of radiomics features with genomic analysis is an emerging area. The hypothesis behind this model of analysis is that the imaging phenotype (-radio), obtained in vivo and in a non-invasive way should reflect the genomic phenotype (-genomics) described by the mRNA profile of the tumor. 

In the last two decades, the number of papers dedicated to radiogenomics in oncology, and in particular in BC studies, has rapidly increased. It is expected that information obtained by such analysis could impact in a better characterization of primary BC thus selecting those patients who will benefit by a specific therapy/surgical treatment. In this section, an overview of the recent literature involving the integration of IFs and gene expression profiles will be provided. Table 1 summarizes the main findings on these radiogenomic studies focused on breast cancer.

Using Cancer Genome Atlas (TCGA) and the corresponding imaging information stored in The Cancer Imaging Archive (TCIA), Mazurowski et al. [67] applied computer vision algorithms for the extraction of 23 IF on 48 BC patients. These IFs were tested for their association with molecular subtypes determined on the basis of genomic analysis. The authors showed that the luminal B subtype is associated with DCE-MRI features of both the tumor and the background parenchyma.

In an approach involving qualitative radiogenomics, Woodard et al. [76] investigated the relationship between semantic features (i.e., BI-RADS lexicon from mammography and MR imaging) and clinically available genomic assay OncotypeDX recurrence risk scores. On a large cohort, the authors suggest that qualitative radiogenomics data indicate that BI-RADS descriptors might be potential imaging biomarkers of BC recurrence risk. 

We have recently proposed a new combinatorial approach combining radiomics with epigenomic miRNA profiles in BC subtypes, generating a new field of research called radiomiRnomics [69]. BC has been molecularly divided into four types, namely Luminal A, luminal B, HER-2 enriched and basal-like. Each of these subtypes demonstrate differences in response to therapy [77,78]. The TCGA and the corresponding imaging information stored in TCIA allowed the development of a computational approach that correlates the phenotype from DCE-MRI with microRNAs (miRNAs), mRNAs, and regulatory networks, developing a radiomiRNomic map in the different BC molecular subtypes. The obtained results have to be validated on an enlarged prospective cohort. 

Using a similar approach, Incoronato et al. [70] evaluated the associations between the deregulation of circulating miRNAs with high diagnostic accuracy for BC and morpho-functional characteristics of the tumour, as assessed in vivo by PET/MRI. Even if the study was performed on a limited cohort and no advanced textural IFs were provided, standard quantification imaging biomarkers from DCE-MRI, DWI-MRI and PET were found to be correlated with circulating miRNAs (miR-143-3b and miR-125b-5p). In particular, the authors used 77 BC patients performing in the same day PET/MRI analysis and blood collection and 78 healthy subjects that were recruited as negative controls. Of the 84 miRNAs identified, the authors found that miR-125b-5p, miR-143-3p, miR-145-5p, miR-100-5p and miR-23a-3p were significantly upregulated in plasma BC samples. A strong correlation was obtained between the expression level of miR-143-3p and the mean initial area under the concentration curve (iAUCmean) and the mean reverse efflux volume transfer constant (Kepmean) in stage II BC, suggesting a possible role for miR-143-3p in the regulation of tumor vascularization. Moreover, a strong correlation was observed between miR-143-3p and the maximum standardized uptake value (SUVmax) at stage II, suggesting that this miRNA is also involved in the control of tumor metabolism. miR-125-5p was inversely correlated with the mean forward volume transfer constant (Ktrans mean) and the proliferation Ki67 index at stage IV. As Ktrans mean is a parameter linked to tumor vascularization, the highest plasma levels of miR-125-5p are predictive of a better prognosis.

The extensive use of genomic profiling techniques has allowed the combination of imaging features with genomic profiles. One of the most recent publications of the combinatorial approach of radiomic features and RNA genomic profile integrates whole transcriptome analysis using RNASeq with 3 T DCE MRI data of patients with BC [71]. This study demonstrated a positive association of IFs extracted from DCE-MRI with multiple replication and proliferation pathways and a negative association with the apoptosis pathway: in particular, the authors claimed that the increased expression of the apoptosis genes was associated with more spherical and less irregular breast tumors. Tumors with increased immune activation appeared to be more confined on imaging, as shown by negative correlation with size features and positive correlation with sphericity. Tumors with immune activation also have the tendency to be more heterogeneous in texture, as shown by a positive correlation with IF Entropy from texture analysis and a negative correlation with IF Energy from texture analysis. The vascular endothelial growth factor (VEGF) pathway and several radiomic phenotypes have been associated, such as the dynamic contrast enhanced MRI, which quantifies changes in the microvascular physiology of tumors [79]. The increased expression of VEGF was positively correlated with the variability of enhancement and negatively correlated with IF homogeneity from texture analysis, supporting the idea that VEGF activation may result in the disorganized formation of “leaky vessels” quantifiable on DCE-MRI [71].

In the Multimodality Analysis and Radiological Guidance in Breast-Conserving Therapy (MARGIN) study, the authors collected 21 imaging features from MRI analysis, condensed into 7 MRI factors (tumor size, tumor shape, initial enhancement, late enhancement, smoothness a of enhancement sharpness, and sharpness variation) and associated these factors with gene expression profiles obtained by RNA sequencing analysis [72]. The study involved 295 patients. The authors found a strong association between proliferation and tumor size. Highly proliferative tumors are those with poor prognosis, and larger tumors have a worse prognosis than smaller ones. Moreover, they found that low initial enhancement, increased smoothness and low sharpness are associated with the expression of ribosomal mRNAs, which is a well-known target of chemotherapeutic agents; increased smoothness of enhancement, smaller tumor size, and an irregular tumor shape are associated with the expression of genes of the extracellular matrix and collagen production, highlighting the role of fibroblast in cancer progression.

In a recently published paper [73], the authors used Dynamic Contrast Enhanced Magnetic Resonance Imaging of 73 patients to identify which of the MRI-derived radiomics features could predict cell invasion in the tumor microenvironment. They found that the size and morphology radiomics features (such as diameter and perimeter) correlate positively with neutrophil abundance, while the abundance of fibroblasts and endothelial cells correlates with kinetic features of the tumor (positive correlation with mean quick WIS, negative correlation with Tumor skewness in Post-contrast2). Also, in this paper, fibroblast involvement in collagen and connective tissue deposits emerges as a predictive factor of invasive ability of the tumor cells, observed both in MRI and CT imaging and in genomic profiling. Moreover, endothelial cells, CD8+ T cells and neutrophils are also predicted using radiomic features, confirming the robustness of radiomic features in reflecting the association with the cell type of CD8+.

In [74] the authors integrated multi-omics molecular data from TCGA (also including miRNA profiles) with MRI imaging data from TCIA for 91 invasive carcinomas. They obtained a strong association between the transcriptional activity of the tumors and four tumor size phenotypes (lesion volume, effective diameter, surface area, and maximum linear size). On the contrary, there is a strong negative association between transcriptional activity and tumor morphological features (margin sharpness and variance of radial gradient histogram), meaning and association between transcription activity and a blurred tumor margin, as a sign of tumor invasion within the surrounding microenvironment. Also, the transcription pathway is positively associated with the irregular tumor shape (irregularity and surface to volume ratio). Regarding the miRNA profile, they found a high association between miRNA expression (in particular miR-128-1 and miR-18a) and tumor size and enhancement texture. This would suggest that miRNAs mediate tumor growth and the heterogeneity of the blood vessel system of the tumor. miRNAs of the cluster miR-17-92 would be associated with the enhancement texture phenotype (including contrast, correlation, difference variance, entropy and maximum correlation coefficient), a sign of the aggressive phenotype of the lesion. On the contrary, Let7b expression is negatively associated with enhancement texture phenotype, possibly due to the tumor suppressor activity of this miRNA. miR-10b expression is associated with the effective diameter of the tumor, acting as a modulator of tumor invasion and metastasis.

Little is known of the relation among long non-coding RNA (lncRNA) expression and the tumor phenotype described by MRI and CT imaging data. Yamamoto and collogues in 2015 [75] performed a radiogenomic analysis of BC patients (n = 70) by dynamic contrast material-enhanced (DCE) MRI and correlated the imaging features with early metastasis and lncRNA expression obtained by next generation sequencing. They found that the enhancing rim fraction (ERF) score is strongly correlated with the early occurrence of metastasis. The ERF score, dividing the patients into low-expressing and high-expressing, allows the identification of eight lncRNAs, of which five (RP11-278, L15.2-001, LINC00511-009, HOTAIR, AC004231.2-001) are strongly, positively associated with the ERF score. These lncRNAs are mainly involved in the control of the cell cycle, cell survival or apoptosis, cellular development, and cell growth.

## 4. Limitations of Radiogenomic Approach

Molecular characterization of BC uses genomic, transcriptomic and proteomic tools, and requires tissue sampling from invasive surgery or tissue biopsy. The bioptic sample may not be representative of the entire lesion, being obtained from a small portion of the heterogeneous lesion of the tumor. That’s the reason for collecting blood withdrawn in order to obtain a profile of possible circulating cancer-associated molecules, such as cfDNA or RNA. Large-scale genomic profiling is not feasible due to high cost, although it would allow the genetic profiling of a big population. Another issue is the high amount of data to be stored, a complex approach for data analysis and interpretation. This severely limits the development of prospective clinical trials of radiogenomics.

At present, there are few databases (mainly TCGA and TCIA) associating imaging data with genomic profiling, and the reduced number of samples in these databases. Often these databases do not contain the immunohistochemistry images to be compared and correlated with all of the other data. Moreover, bioptic samples in which genetic/epigenetic analyses have been performed might not be representative of the entire tumor, as seen by imaging techniques. The ideal database should contain radiomic feature data, genomic (mutations, copy number variations, etc.), epigenomic (non-coding RNAs), proteomic, metabolomic, transcriptomics, and immunohistochemical information altogether.

Furthermore, an effort to standardize the imaging and biochemical techniques of analyses is needed to define stable and reproducible radiogenomics biomarkers.

Despite this, the interest of the radiogenomic approach is expected to stimulate the development of international standardized perspective studies for recruitment of a large amount of data to support clinicians in the personalized diagnosis of BC. Standardization of the non-coding RNA profiles as well as of the technique and the machine used for imaging acquisition and the selection of imaging features are still challenges. Finally, for the generalization of the results, a large cohort and biological sample collection is necessary in order to have reliable results.

Figure 2 summarizes the limitations and challenges of radiogenomics.

## 5. Future Perspectives

Radiogenomics is the combination of genomic and radiomic data. This new approach could become a promising tool to increase precision in diagnosis, providing important in vivo information on BC behaviour and development. Indeed, precision medicine could be optimized considering the genotypic and phenotypic characteristics of a tumor. The approach of precision medicine is based on system biology, which integrates mathematical modelling and cellular biology, genomics, transcriptomics, proteomics, metabolomics and epigenomics (Figure 3). Radiomics, extracting large volumes of quantitative data from digital images and matching them with clinical and patient data, could help in the development of new radiogenomics approaches integrating this information into public shared databases containing molecular profiles. Radiogenomics may provide imaging and genetic information voxel-by-voxel for a complete, heterogeneous tumor. 

## Figures and Tables

**Figure 1 mps-05-00078-f001:**
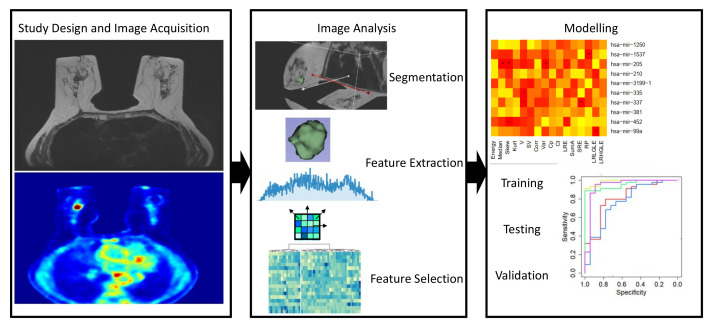
The radiomic workflow.

**Figure 2 mps-05-00078-f002:**
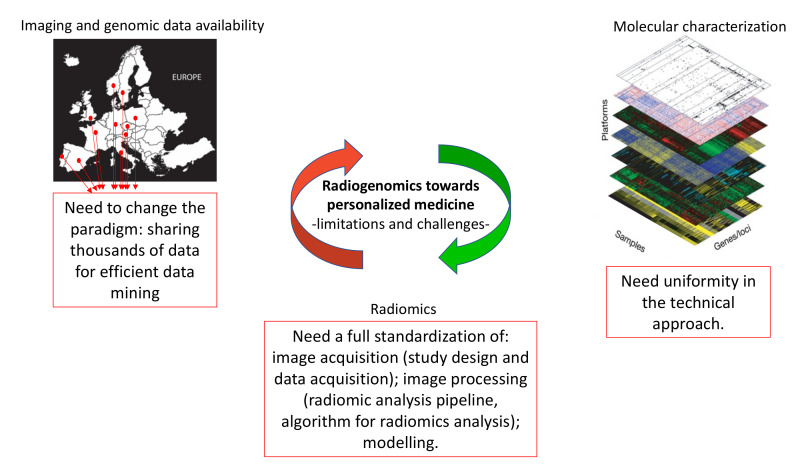
Limitations and challenges of radiogenomic approach.

**Figure 3 mps-05-00078-f003:**
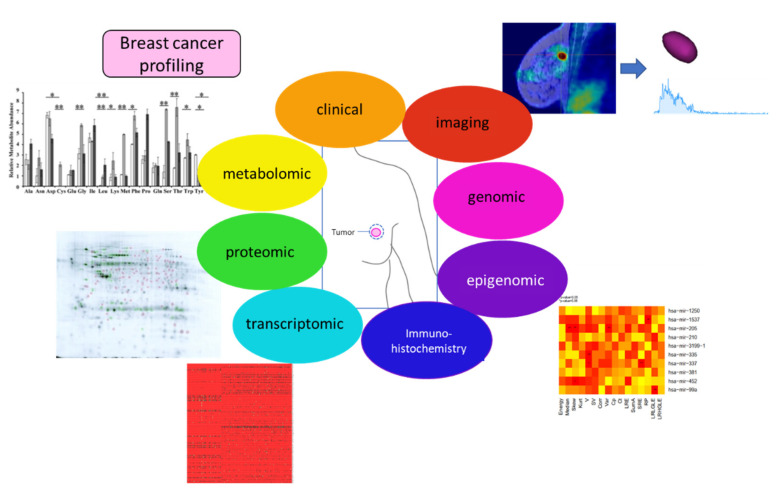
Integrative approaches for a better description of BC early biomarkers. Images were taken and modified from [69,80].

**Table 1 mps-05-00078-t001:** Summary of literature findings on radiogenomics.

Imaging	Aim	BC Patients	Data Source	IFs	Findings	Ref.
DCE-MRI	molecular subtype (determined on the basis of genomic analysis) vs. IFs	N = 48	TCIA-TCGA	Morphological IFs, Local intensity IFs (from kinetics) GLCM IFs	There is an association between dynamic contrast material–enhancement IF that quantifies the relationship between lesion enhancement and background parenchymal enhancement and luminal B subtype	[67]
Mammography and MRI	IFs vs. Oncotype DX Test Recurrence Score	N = 408	Retrospective in-house clinical protocol	Semantic IFs	Semantic IFs from mammography and MRI can be used for imaging biomarkers of breast cancer recurrence risk	[68]
DCE-MRI	IFs vs. miRNAs, mRNAs, and regulatory networks	N = 37	TCIA-TCGA	Morphological IFs, Histogram intensity IFs, GLCM IFs, GLRLM IFs	A radiomiRNomic signature including both miRNAs and imaging features have better classification power of Luminal A versus the different BC subtypes than using miRNAs or imaging alone	[69]
PET/MRI	IFs vs. circulating miRNAs	N = 77	Prospective in-house clinical protocol	Morphological and Local intensity IFs	Different Local intensity IFs have a correlation with miRNAs expression, showing potential for risk stratification of BC and to improve diagnostic accuracy	[70]
DCE-MRI	IFs vs. and RNA genomic profile	N = 47	Retrospective in-house clinical protocol	Morphological IFs, Local intensity IFs (from kinetics) GLCM IFs,	Several molecular pathways related to replication, proliferation, apoptosis, immune system regulation and extracellular signalling have a robust association to IFs	[71]
DCE-MRI	IFs vs. gene expression levels from RNA sequencing	N = 295	Prospective in-house clinical protocol	Morphological IFs, Local intensity IFs (from kinetics)	DCE-MRI phenotypes are related to underlying molecular biology revealed by using RNA sequencing	[72]
DCE-MRI	IFs for prediction of cell invasion in the tumor microenvironment	N = 73	TCIA-TCGA	Morphological IFs, Histogram intensity IFs, GLCM IFs, GLRLM IFs, GLSZM IFs,	Univariate correlations of IFs and abundance of fibroblasts. Multivariate models with AUCs ranging from 0.5 to 0.68 for the multiple cell type invasion predictions	[73]
DCE-MRI	IFs vs. DNA mutation, miRNA expression, protein expression, pathway gene expression and copy number variation	N = 91	TCIA-TCGA	Morphological IFs, Local intensity IFs (from kinetics) GLCM IFs	MRI is a potential non-invasive approach to probe the cancer molecular status, since several transcriptional activities of various genetic pathways were positively associated with different IFs	[74]
DCE-MRI	IFs vs. lncRNA expression and MFS	N = 70		Morphological IFs, Local intensity IFs (from kinetics), Histogram intensity IFs	5 lncRNAs, involved in the control of cell cycle, cell survival or apoptosis, cellular development, and cell growth, are associated with IFs	[75]

IFs Imaging Features; TCIA Tumor Cancer Imaging Archive; TCGA Tumor Cancer Genomic Archive; MRI Magnetic Resonance Imaging; DCE-MRI Dynamic Contrast Enhanced MRI; GLCM Grey Level Co-Occurrence Matrix GLRLM Grey Level Run Length Matrix; GLSZM Gray Level Size Zone Matrix; GLDM Gray Level Dependence Matrix; NGTDM Neighboring Gray Tone Difference; Positron Emission Tomography/Magnetic Resonance Imaging (PET/MRI); miRNAs non-coding RNA molecules; lncRNA long noncoding RNA; MFS metastasis-free survival.

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
