# Peer review of "Radiogenomics, Breast Cancer Diagnosis and Characterization: Current Status and Future Directions"

_mps, 2022, doi:10.3390/mps5050078_

Round 1
Reviewer 1 Report
General Comment:
This work reviews the radiogenomics approaches for the diagnosis of breast cancer. The literature is analyzed to identify the approaches which combine imaging and biochemical/biological data.
Specific comments throughout the paper:
The"abstract should be a total of about 200 words maximum. The abstract should be a single paragraph and should follow the style of structured abstracts, but without headings". Please see
https://www.mdpi.com/journal/mps/instructions#preparation
1. Introduction
Line 29: "pandemia" should be "pandemic" - Please fix.
Line 32: missing space
Lines 36-40: Missing reference(s)
Line 45: Why using the "? The right and exact word is features. Please correct. Also check line 49.
Lines 46-49: Missing ref.
Line 58: Too many square parentheses.
2. Biomarkers for BC
2.1
Lines 64-66: Lack of referencing for this definition.
Lines 68-74: No references are given for the liquid biopsy. The quality of the referencing and of the Introduction must be improved.
Line 79: "ctDNA" is not defined.
Line 80: PCR and other acronyms should be defined. The understanding and readibilty should be improved.
Line 117: extra space. Check the editing.
2.2
Lines 122-126: Missing ref
Lines 127-139: This part must be extended. For improving this section, please read
10.1002/jmri.26518
10.1259/bjr.20190948
The technical definition, the computation, and the "engineering" part is not faced in a proper way. The value of and the interest in this paper could be increased by widening the audience.
Line 143: Please provide a figure describing graphically the radiomic workflow.
Line 151: Please fix.
Line 172: The author have not inserted properly the right reference.
Line 207, 223: Please revise accordingly to previous comments.
3. Radiogenomics: combining molecular and imaging biomarkers for BC characterization
Line 227: IFs not defined as acronym
A summary table for the literature analysis would be a plus and added value to the manuscript. The readership would severely benefit from this.
Line 294: VEGF not defined as acronym
Line 299: A list of the imaging features would be appreciated
4. Limitations of radiogenomic approach
This section is appreciated. A summary table/figure would increase the quality of the work.
5. Future perspectives
The section numbering must be revised.
Author Response
Review of the article : “Radiogenomics and breast cancer diagnosis: current status and future directions.”
We thank the reviewer for the observations and suggestions. We modified the text in accordance with the requests. We replied point-by-point to each reviewer’s observation, indicating the lines in the revised manuscript where we modified the text.
General Comment:
This work reviews the radiogenomics approaches for the diagnosis of breast cancer. The literature is analyzed to identify the approaches which combine imaging and biochemical/biological data.
Specific comments throughout the paper:
The"abstract should be a total of about 200 words maximum. The abstract should be a single paragraph and should follow the style of structured abstracts, but without headings". Please see
https://www.mdpi.com/journal/mps/instructions#preparation
We thank the reviewer for this observation. We modified the abstract accordingly (See Abstract section line 9-22 of the revised manuscript).
- Introduction
Line 29: "pandemia" should be "pandemic" - Please fix.
Fixed as suggested by the reviewer.
Line 32: missing space
Fixed as suggested by the reviewer.
Lines 36-40: Missing reference(s)
Fixed as suggested by the reviewer: (references 5-9 added).
Line 45: Why using the "? The right and exact word is features. Please correct. Also check line 49.
Fixed as suggested by the reviewer.
Lines 46-49: Missing ref.
Fixed as suggested by the reviewer (reference 11 added).
Line 58: Too many square parentheses.
Fixed as suggested by the reviewer.
- Biomarkers for BC
2.1 Lines 64-66: Lack of referencing for this definition.
Fixed as suggested by the reviewer (please see ref. 16).
Lines 68-74: No references are given for the liquid biopsy. The quality of the referencing and of the Introduction must be improved.
Fixed as suggested by the reviewer (please see ref. 21).
Line 79: "ctDNA" is not defined.
Fixed as suggested by the reviewer (added in line 98).
Line 80: PCR and other acronyms should be defined. The understanding and readibilty should be improved.
Fixed as suggested by the reviewer (please see line 107).
Line 117: extra space. Check the editing.
Fixed as suggested by the reviewer.
2.2 2 Lines 122-126: Missing ref
Fixed as suggested by the reviewer (Please see ref. 32).
Lines 127-139: This part must be extended. For improving this section, please read
10.1002/jmri.26518
10.1259/bjr.20190948
The technical definition, the computation, and the "engineering" part is not faced in a proper way. The value of and the interest in this paper could be increased by widening the audience.
Line 143: Please provide a figure describing graphically the radiomic workflow.
We thank the reviewer for the observations. In the revised manuscript, we have extended this part including the description of methodological aspects of the radiomics workflow, inserting proper references and a figure for illustrating the radiomic pipeline (see Section 2 “Biomarkers for BC”, Subsection 2.2 ” BC imaging biomarkers: from standard quantification to radiomics”, Paragraphs from 3 to 7, and Figure 1).
Line 151: Please fix.
Fixed as suggested by the reviewer.
Line 172: The authors have not inserted properly the right reference.
Fixed by correcting the reference number.
Line 207, 223: Please revise accordingly to previous comments.
Fixed as suggested by the reviewer.
- Radiogenomics: combining molecular and imaging biomarkers for BC characterization
Line 227: IFs not defined as acronym
Fixed as suggested by the reviewer.
Line 294: VEGF not defined as acronym
Fixed as suggested by the reviewer.
A summary table for the literature analysis would be a plus and added value to the manuscript. The readership would severely benefit from this.
Line 299: A list of the imaging features would be appreciated
We thank the reviewer for the suggestion. We provided a summary table (see Table 1 in Section 3 “Radiogenomics: combining molecular and imaging biomarkers for BC characterization”) for the considered literature. Since each considered published study, evaluated different imaging features, we have included in Table 1 also a column indicating, for each study, the classes of considered imaging features on the basis of the classification indicated by the IBSI initiative.
- Limitations of radiogenomic approach
This section is appreciated. A summary table/figure would increase the quality of the work.
We thank the reviewer for the suggestion. We have added Figure 2 highlighting limitations of the radiogenomic approach.
- Future perspectives
The section numbering must be revised.
Fixed as suggested by the reviewer.

Reviewer 2 Report
This review presents a short overview about molecular, radiomic and radiogenomic analyses adopted for breast cancer diagnosis, with a focus on approaches related to RNA analysis.
1. In my opinion, the largest limitation of this work is represented by the small innovation introduced in the manuscript, since several other recent reviews are focused on the same topic (see for example Pinker 2018 https://doi.org/10.1148/radiol.2018172171, Grimm 2020 https://doi.org/10.1016/j.acra.2019.09.012, and Darvish 2022 https://doi.org/10.1186/s43042-022-00310-z). For this reason, I believe that the authors should try to better indicate which is the novelty that they want to present here with respect to the literature.
2. I suggest the authors to improve the description about molecular biomarkers. An initial definition of genomics, transcriptomics, proteomics, etc could help the reader to better understand the different approaches and where the listed molecules (cfDNA, CTCs, and ncRNA) can be located. Moreover, molecular subtypes of breast cancer determined by genomic analysis should be defined in this section, for a better understanding of the next radiogenomics section.
3. In the title and introduction, the focus of this review is on breast cancer diagnosis; however, some of the described techniques/works are not used for diagnosis. For example, in the radiomic section, MRI is mostly adopted before treatment to build prediction models for tumor response. Please, clarify this point.
4. Finally, some minor English revisions and reference checks are needed. For example: page 2, line 83: the reference in brackets should be correctly formatted; page 4, lines 154-158: please, check and rephrase the sentence; page 4, line 172: missing reference.
Reviewer 3 Report
Dear Authors/Editors.
In my opinion, this review gives a good introduction to the topic for this special issue. The aim of the authors is to provide an overview of the latest, most interesting developments in the field of radiomics and radiogenomics for breast cancer. The review is structured into the topics 1. biochemical markers, e.g. for the detection of circulating tumor DNA or circulating tumor cells, 2. radiomics in imaging, e.g. for the detection of lymph node metastases before therapy, 3. radiogenomics, the combination of molecular markers and radiomics, 4. Limitations of Radiogenomics and 5. Future Perspectives.
The review is very well done and I find it very easy to accept the review for the publication.
Round 2
Reviewer 1 Report
The abstract was revised.
Typos and errors were fixed, so the quality of the paper improved.
The introduction section was strengthened by improving the referencing. Also, the style and editing was revised thoroughly.
I must thank the authors for improving the engineering part related to the imaging. The methodological rigour and value of this work significantly improved.
Thanks for providing Tab. 1. It requires some editing and re-style.
The references style needs to be updated and revised.
Reviewer 2 Report
I want to thank the authors for their efforts in addressing all my requests and those from the other reviewers. Now the manuscript is significantly improved both in the originality and quality of the review.
Before accepting the work, I ask the authors to perform a final English spell-check and a careful references revision. I list some changes/errors in the following points:
- page 2, line 46-47: the sentence is not clear; "from extracting quantitative features from the images" could be modified as "to extract quantitative features from the images"?
- page 2 line 51: remove comma after "The new paradigm of radiomics"
- page 2 line 65-70: to increase the readability of this last paragraph, I suggest to use shorter sentences (i.e. split this last sentence in at least two parts)
- page 5 line 205: change "produce" as "produces"
- page 5 line 221: please check references currently indicated as A13
- page 6 lines 256-257: change "acquisition techniques (such as elastography, Doppler, or contrast-enhanced US (CEUS) can" as "acquisition techniques, such as elastography, Doppler, or contrast-enhanced US (CEUS), can"
- Table 1: please check references in the square brackets in the first three rows
- Figure 2: please try to incerase the text size in the boxes, it is currently too small to be readable.
